# Structural basis for autoinhibition by the dephosphorylated regulatory domain of Ycf1

Nitesh Kumar Khandelwal[1,2] & Thomas M. Tomasiak ⊕ [1] ✉

Yeast Cadmium Factor 1 (Ycf1) sequesters glutathione and glutathione-heavy metal conjugates into yeast vacuoles as a cellular detoxification mechanism. Ycf1 belongs to the C subfamily of ATP Binding Cassette (ABC) transporters characterized by long flexible linkers, notably the regulatory domain (R-domain). R-domain phosphorylation is necessary for activity, whereas dephosphorylation induces autoinhibition through an undefined mechanism. Because of its transient and dynamic nature, no structure of the dephosphorylated Ycf1 exists, limiting understanding of this R-domain regulation. Here, we capture the dephosphorylated Ycf1 using cryo-EM and show that the unphosphorylated R-domain indeed forms an ordered structure with an unexpected hairpin topology bound within the Ycf1 substrate cavity. This architecture and binding mode resemble that of a viral peptide inhibitor of an ABC transporter and the secreted bacterial WXG peptide toxins. We further reveal the subset of phosphorylation sites within the hairpin turn that drive the reorganization of the R-domain conformation, suggesting a mechanism for Ycf1 activation by phosphorylation-dependent release of R-domain mediated autoinhibition.

Yeast Cadmium Factor 1 (Ycf1) belongs to the ATP Binding Cassette (ABC) transporter family and sequesters glutathione and glutathione heavy metal conjugates to the vacuole to detoxify reactive oxygen species and heavy metals ($Cd^{2+}$, $Se^{2+}$, $Hg^{2+}$, $As^{2+}$)[1–3]. In eukaryotes, ABC transporters are divided into five subfamilies (A, B, C, D, and G) with different topologies built around a core architecture made of two nucleotide-binding domains (NBDs) where ATP is hydrolyzed and two transmembrane domains (TMDs) through which substrate is transported. Ycf1 belongs to the C subfamily (termed ABCC), which, in addition to the core architecture, includes several evolutionary additions, including an additional TMD (called TMD0), a lasso domain that links TMD0 and TMD1, and a flexible linker domain called the regulatory domain (R-domain) that connects NBD1 and TMD2.

R-domains play prominent roles in transporter activation and have unique properties, making them hubs of signaling integration[4]. They are typically large (~60–200 amino acids), and their phosphorylation is indispensable for activity in Ycf1 and the cystic fibrosis transmembrane conductance regulator (CFTR)[5–7], a related chloride transporter that causes cystic fibrosis when mutated[8] and can also transport glutathione[9]. R-domains connect NBD1 to TMD2, placing them near several key allosteric and catalytic elements, such as ATP binding sites. In Ycf1, mutation of two R-domain sites (S908 and T911) results in loss of cell survival in the presence of cadmium, a Ycf1 substrate. CFTR contains a more extensive R-domain with six activity-driving phosphosites and 20 disease-associated mutation site[8,10]. In CFTR, activity also depends on R-domain phosphorylation driven by interactions with Protein Kinase A (PKA) and other kinases such as PKC and it serves as the site of several protein-protein interactions[4]. In Multidrug Resistance Protein 1 (MRP1), another glutathione and glutathione conjugate ABCC transporter[11] and proposed ortholog of Ycf1[12], the Rdomain also controls relative substrate selectivity for the heavy metals selenium or arsenic[13].

Conflicting reports detail how the R-domain structure transitions between phosphorylated and dephosphorylated states. It has been proposed that the R-domain is disordered mainly because of low sequence complexity dominated by repeats of negatively charged

[1]Department of Chemistry and Biochemistry, University of Arizona, Tucson, AZ 85721, USA. [2]Present address: Department of Biochemistry and Biophysics, University of California – San Francisco, San Francisco, CA 94158, USA. ✉e-mail: tomasiak@arizona.edu

amino acids thought to have derived from an intronic evolutionary origin in CFTR[14]. This conclusion is supported by several biophysical investigations in isolated R-domains by NMR[15], circular dichroism[16], and protease accessibility[16], that show little secondary structure and suggest substantial intrinsic disorder. They also imply that the R-domain mainly resides between NBD1 and NBD2 when unphosphorylated, thereby poised to play an autoinhibitory role[17]. In this model, the R-domain mobility appears to become even more diffuse upon phosphorylation and is subsequently released from its position between the NBDs to relieve autoinhibition and permit activity.

In contrast to these findings from isolated segments, investigations in full-length transporters, including those performed by cryo-EM, provide evidence that more well-ordered intermediates of the R-domain might exist. For example, although the R-domain is completely unresolved in cryo-EM structures of dephosphorylated CFTR, they do show diffuse electron potential density for the R-domain oriented between both halves of CFTR[18]. On the other hand, phosphorylated CFTR structures show limited R-domain segments built only as Cα traces because of poor resolvability; nevertheless, this data suggests interactions along the periphery of NBD1. More recently, high-resolution Ycf1 cryo-EM structures have clarified this region and revealed the topology of a well-resolved R-domain and its residues bound to NBD1[5,19]. In addition, a cryo-EM structure of phosphorylated Ycf1 provided the definitive maps with density corresponding to a trio of phosphorylated residues (S908, T911, and S914) that directly engage basic surfaces within this region[5]. Finally, surface accessibility studies on full-length CFTR[20] and Ycf1[19] suggest that the dephosphorylated R-domain engages a significant portion of the TMDs, while the phosphorylated R-domain only engages NBD1. This work is consistent with earlier models that the R-domain may directly block the conductance pore[21]. Altogether, these structures point to an R-domain architecture far more structured than previously alluded to by biophysical investigations.

Here, we report the cryo-EM structure of Ycf1 in a dephosphorylated state that captures a pore-occluded state with key allosteric and orthosteric interactions. Changes in substrate-induced ATPase acceleration, cell survival on toxic heavy metal cadmium, and accessibility to proteases support an updated model for the dephosphorylated R-domain inhibitory mechanism as well as insights into ABCC family transporter activation.

## Results

### Dephosphorylated Ycf1 resembles phosphorylated IFwide but lacks the R-domain around NBD1

Many ABCC family transporter family members are regulated by dynamic phosphorylation of an intrinsically disordered R-domain[22]. Our previous cryo-EM structure of Ycf1 revealed a partially structured conformation of the phosphorylated R-domain engaged with NBD1 that, upon dephosphorylation with alkaline phosphatase, drastically lowered basal ATPase activity and caused changes to the Ycf1 architecture[5]. Here, we sought to determine the structural changes in the dephosphorylated Ycf1 E1435Q mutant, expressed and purified from the *Saccharomyces cerevisiae* DSY5 strain. This mutant was chosen, as in our previous study, to limit heterogeneity and has successfully been used in the structure determination of CFTR, MRP1, BmrCD, and other landmark ABC transporter structures[18,23,24]. We generated the dephosphorylated state by treating purified phosphorylated Ycf1 with alkaline phosphatase, followed by a subsequent round of size exclusion purification to yield a highly homogenous dephosphorylated sample (Fig. 1A - top) verified by phosphoprotein staining (Fig. 1A − bottom).

The homogenous sample enabled cryo-EM structure determination of the dephosphorylated Ycf-E1435Q to an overall final resolution of 3.11 Å (Fig. 1B−D and Figure S1), which is higher resolution than any previously published phosphorylated structures (ranging from 3.2- 4.0 Å[5,19]. Surprisingly, dephosphorylation of Ycf1 leads to a complete loss

of the R-domain density at its interaction site on NBD1 (Fig. 1B−D) previously observed in the phosphorylated structures (Fig. 2). Instead, considerable additional electron potential density was visible within the substrate-binding cavity between TMD1 and TMD2. This density displayed several continuous segments that allowed for the structural assignment of the C-terminal half of the R-domain (residues 890-935) continuous with TMD2 (Fig. 3A, see Methods for details of structure assignment).

Apart from the substantive gross R domain rearrangement, the dephosphorylated Ycf1 architecture remains almost unchanged from the IFwide conformation of phosphorylated Ycf1 (Fig. 2)[5]. We could successfully build all 17 transmembrane helices (TMHs), the lasso region, both NBDs and much of the R domain in its autoinhibited conformation (Fig. 1B, C and Figure S2). Most of the Ycf1 structure was also of higher quality (as judged by map quality and correlation coefficient) than previous phosphorylated structures, as reflected by the higher resolution (Figure S1B).

### Dephosphorylation leads to R-domain occlusion of the substrate cavity

In our structure, the R-domain directly interacts with the substrate binding cavity (Fig. 3A). At this site, it adopts an unexpected helix-strand hairpin motif constructed of an α-helix (residues 902-912) with pronounced corresponding density in the cryo-EM map. This region encompasses three primary phosphorylation sites on Ycf1 (S903, S908, T911), including the RRAS motif. Preceding this helix, the R-domain helix is capped by a loop packed against a strand (residues 890-898) that packs tightly against the R-domain helix. The R-domain eventually connects to a helix leading to TMD2 (Fig. 3A−C). The architecture of the visible region of secondary structure in the R-domain closely matches that of NMR investigation of the CFTR R-domain[15] (Figure S3), which proposes an α-helical conformation of residues 896-913 that is especially pronounced upon dephosphorylation. Similarly, this helix is completely unwound in our phosphorylated Ycf1 structure[5].

The tightly packed R-domain buries ~1640 Å³ of surface area (Fig. 3A) against the Ycf1 binding cavity. This interaction consists of extensive hydrophobic contacts and several electrostatic interactions. Within the R-domain, the hairpin buries several pairs of hydrophobic residues internally, including Ala907 and Ala910, against Phe896 and Leu894 (Fig. 3C, D and Figure S3B). These residues, along with Leu904, pack against a substantial hydrophobic pocket formed primarily along TM6 and comprised of residues Phe576, Met579, Ile580, Met583, and Val584. Additional contributions include residues from TM3 (Trp422, Leu425) TM4 (Met466), and TM5 (Phe536). A second hydrophobic patch connects to the first by packing Leu912 and the γ-methyl of Thr911 against Met466 A hydrophobic cluster comprised of Ile915 and Phe917 from the R-domain packed against Tyr1058 and Met 477 is also observed.

In addition to the buried hydrophobic area, several key charged-charged interactions are made (Fig. 3B) against an exceptionally broad band of positively charged residues from both sides of the R-domain hairpin. The carboxylate side chains of a pair of R-domain acidic residues in the hairpin (Asp 893 and Asp895), engage with 3 basic residues (Arg1174, Arg1228, and Arg1118) and D909 binds to Arg1066 (Fig. 3B). Following the hairpin. Asp916 interacts with Lys1059, Glu 930 interacts with a pair of basic residues (Arg1112 and Arg1115). Finally, several hydrogen bonds complement the charge-charge interactions and include Asn892-Asp418, Arg905-Ser533, Arg906-Asn1224, Asp909-Gln1070, Asn898 and Ser899 to Gln432, and Thr911 to Lys470.

Finally, the interactions in the substrate cavity accompany dephosphorylated R-domain residue Asp920 contacts with a critical allosteric junction residue R1058 of the "GRD" motif of ICL-1 (Fig. 3D), which couples with the X-loop of NBD1 to communicate ATP binding[25].

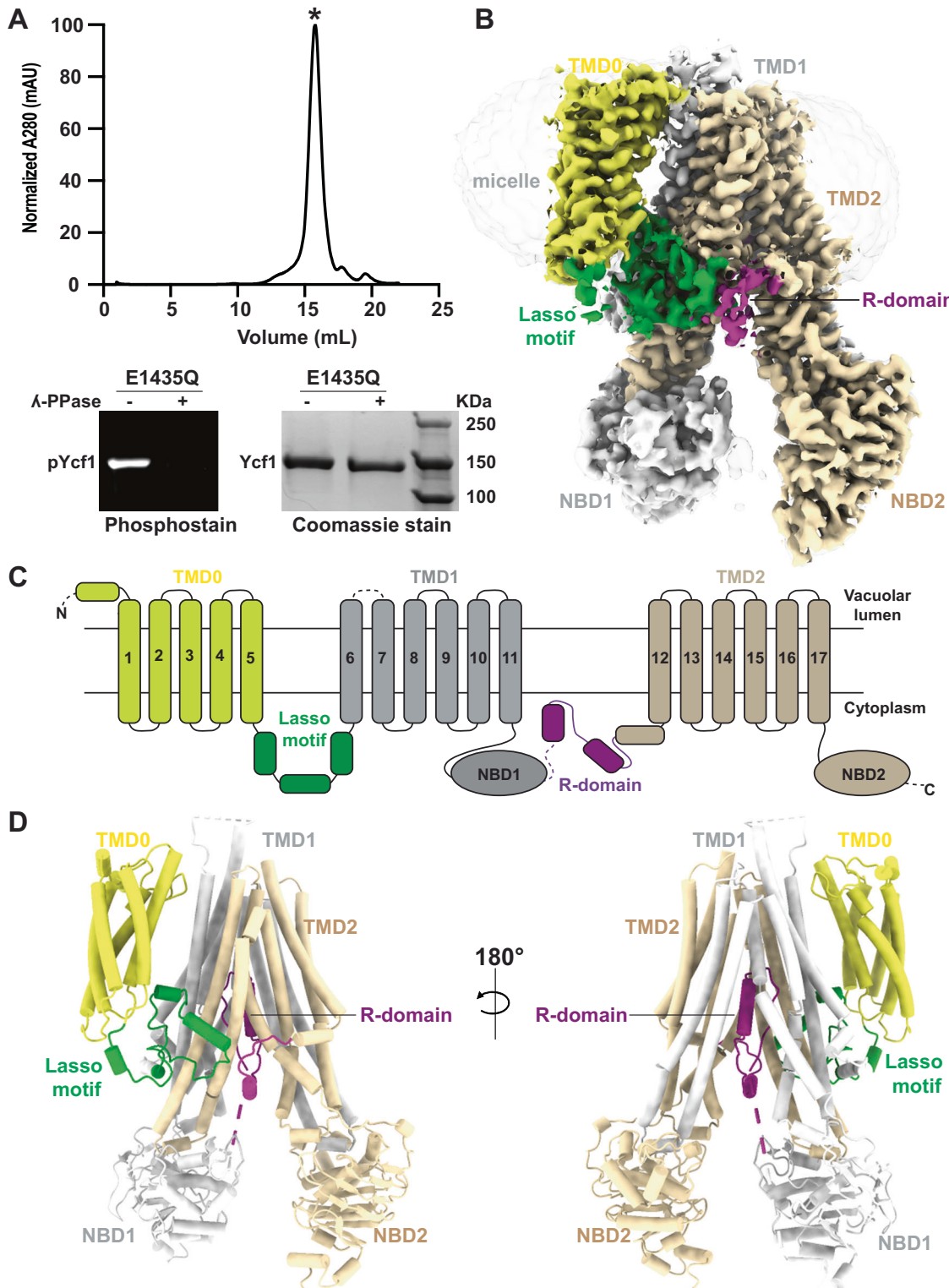

**Fig. 1 | Structure determination of dephosphorylated Ycf1. A** Size exclusion chromatography profile of dephosphorylated Ycf1. This trace is representative of two replicates. Source data are provided as a Source Data file. **B** Cryo-EM density of the dephosphorylated Ycf1 structure. TMD0 is colored yellow, TMD1 and NBD1 are colored white, TMD2 and NBD2 are colored wheat, the lasso domains are colored green, and the regulatory domain (R-domain) is purple. **C** Topology cartoon illustrating the domain architecture of Ycf1 with residue numbers at domain boundaries indicated. **D** Cartoon of the dephosphorylated Ycf1 structure.

## The dephosphorylated R-domain interaction resembles MRP1-substrate and TAP1/TAP2-inhibitor interactions

The altered position of the dephosphorylated R-domain occludes the substrate binding site through an extensive interaction network described above, reducing the cavity volume from 6,464 Å³ in the phosphorylated state to 763 Å³ (Fig. 4A, B). Although a substrate-bound state of Ycf1 is unavailable, the positioning of the R-domain mimics the binding mode of the glutathione moiety of the substrate Leukotriene C4 (LTC4) in the structure of bovine MRP1, a close functional homolog of Ycf1. The residues in MRP1 that bind the glutathione

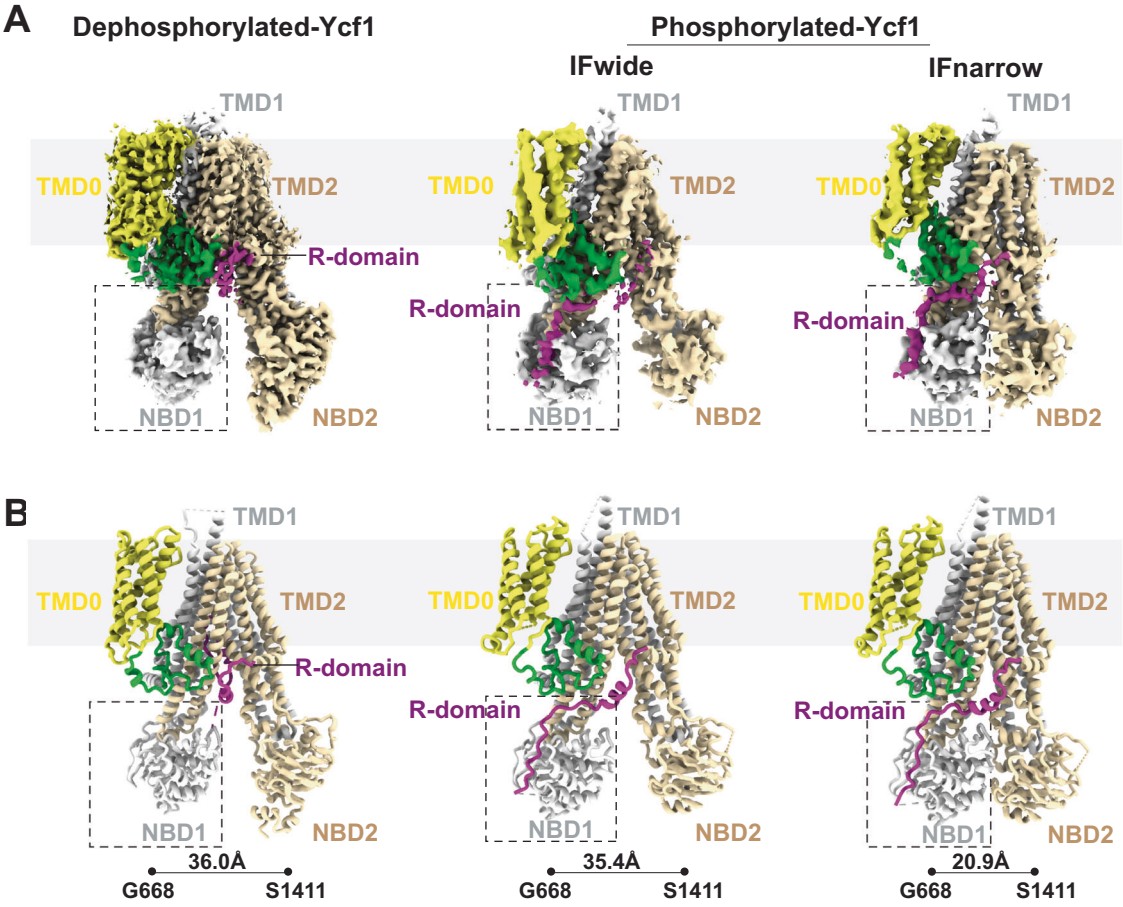

**Fig. 2 | Structure comparison of Ycf1 in dephosphorylated and phosphorylated states. A** Cryo-EM density and (**B**) model of dephosphorylated and phosphorylated Ycf1 structures IFwide (PDBID: 7M69) and IFnarrow (PDBID:7M68) colored as in Fig. 1. The area of missing density for the R-domain in dephosphorylated state around NBD is highlighted with a hashed box.

moiety are conserved in Ycf1 and adopt the same spatial positions in the Ycf1 structure (Fig. 4C). Several residues of the substrate cavity in contact with the R-domain are conserved in the two structures (Fig. 4D, E), including electrostatic interactions with Arg1174 from TM16, and Arg1228 of TM17 (Fig. 4D), which reflects the general electronegative quality of both substrates (glutathione or LTC4). In addition, several hydrophobic sites, such as the Phe367, Trp422, and Phe576, form hydrophobic interactions with several conserved R-domain elements.

Strikingly, alignment of the R-domain structure with the previous structure of the peptide-presenting antigen transporter TAP1/TAP2 bound to an inhibitor from herpes simplex virus[26] also reveals unexpected similarities and structural overlap (Figure S4). The IC47 peptide also exhibits a hairpin-type architecture, with the difference being two helices in IC47, that displays a highly similar set of contacts throughout the substrate cavity. In TAP1/TAP2, these are formed by Trp3, Met7, Phe11, Tyr22, Val25, and Ile29 and are stabilized by a large hydrophobic pocket. At the hairpin connection in TAP1/TAP2, a sizeable basic pocket resembling that observed in Ycf1 forms and is comprised of Arg381 (Tap2), Arg380 (Tap2), and His384 (Tap2).

### Phosphorylation of S903 and S908 on the R-domain helix mainly drives activity

The structural results observed are conducive to a state that blocks enzymatic activity. Basal ATPase activity is strongly inhibited by dephosphorylated Ycf1 and by R-domain mutants, as previously

shown[5,6]. Furthermore, GSSG-dependent stimulation of ATPase activity observed in WT Ycf1 is completely lost in both dephosphorylated Ycf1 and the S908A R-domain mutant (Fig. 5A). Interestingly, the S903A mutation does affect GSSG-potentiated ATPase activity at 8 μM ($EC_{50}$) concentration, whereas partial induction is observed in the T911A variant (Fig. 5A). These results suggest that either 1) glutathione binding is blocked; thus, GSSG cannot induce activity in these mutants, or 2) glutathione binding is not allosterically communicated with the ATPase sites in these mutants, causing loss of GSSG-potentiated ATPase activity.

To measure the overall physiological effect of dephosphorylation on substrate translocation, we performed a yeast survival assay in which Ycf1-dependent transport is necessary for yeast growth on media containing 150 μM $CdCl_2$, a toxic heavy metal substrate of Ycf1. Since inducing a native dephosphorylated state is unattainable in vivo without other widespread cellular changes, this assay is a suitable readout of endogenous transporter function. Consistent with results from ATPase assays and from previously reported cell survival results for S908 and T911[6], the S908A and S903A mutants showed the most significant deficiency in survival (Fig. 5B). The phosphorylation site mutants S869A, S878A, T911A, and S914A all showed no reduction in (or slight elevation of) survival in a manner, again, consistent with ATPase assay results, leading us to conclude that R-domain driven changes in the ATPase rate are responsible for changes in cadmium detoxification.

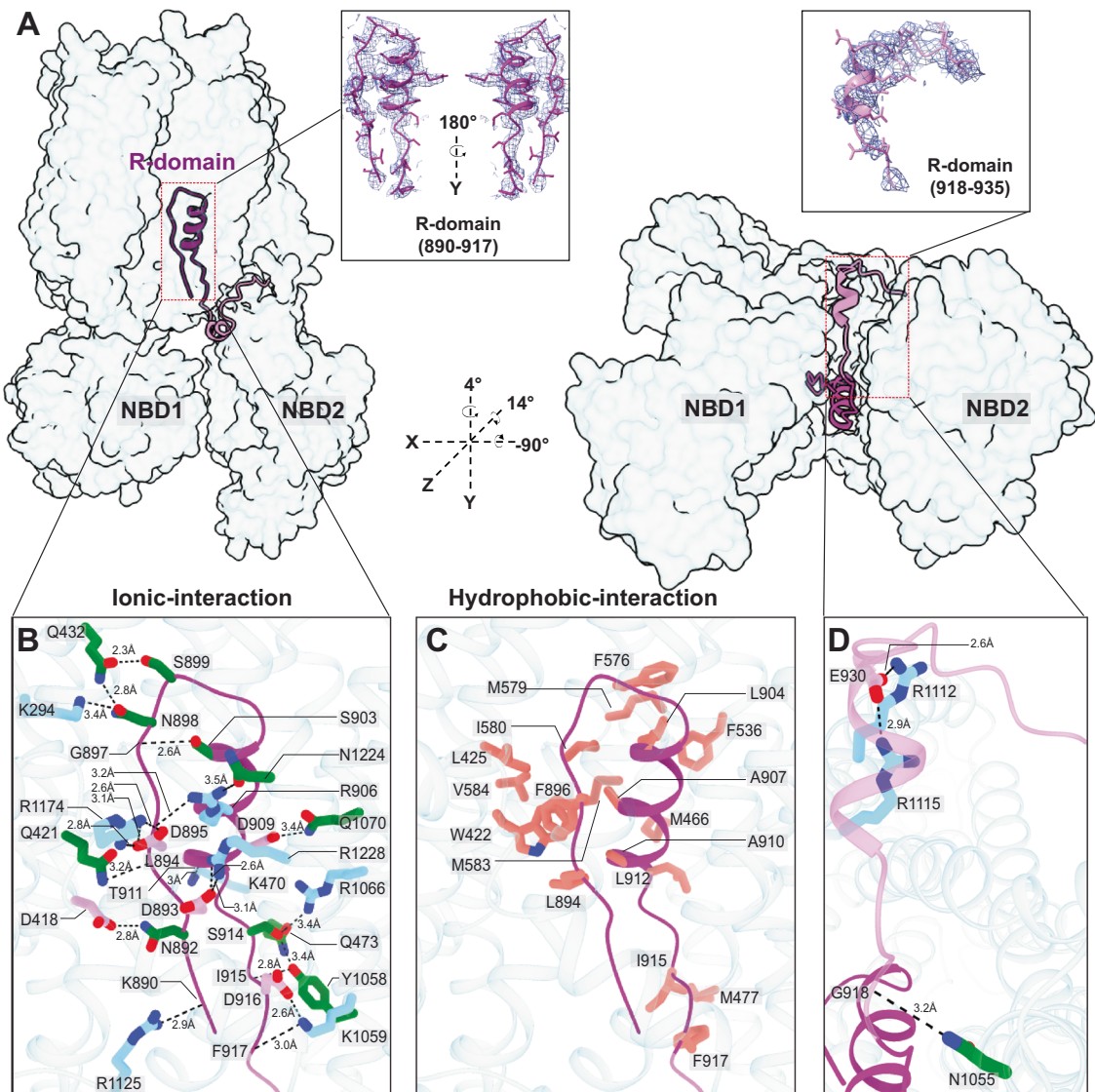

**Fig. 3 | Representation of the R-domain in the Ycf1 substrate cavity. A** A space-filling model of Ycf1 with a transverse section of the TMD removed to show the R-domain from a side view (left) and bottom view of the NBDs (right). **B** Electrostatic interactions between the R-domain and the substrate binding cavity. The R-domain backbone is colored purple, with interacting residues and the Ycf1 backbone colored white. Amino acid side chains interacting between the R-domain and Ycf1 are colored based on physical amino acid properties (Hydrophobic residues are colored orange, positively charged amino residues are in cyan, negatively charged residues are in pink, and polar-uncharged residues colored green. **C** Hydrophobic interactions with the same color scheme as in (**B**). Detailed view of R-domain hydrophobic interactions with the TMD region of Ycf1 and. **D** the C-terminal region of the R-domain.

## R-domain architecture in the S908A variant most closely mimics dephosphorylated Ycf1

We employed a limited proteolysis protection assay to interrogate changes in the R-domain location and structure in dephosphorylated Ycf1 during glutathione stimulation (Fig. 5C). Previously, we observed that dephosphorylated Ycf1 leads to a loss of proteolytic fragments corresponding to the R-domain[5]. Of the mutants with nearly complete activity loss, the S908A mutant displayed a digestion pattern most similar to the dephosphorylated form of Ycf1 presented in our previously published study[5]. Missing bands in the digest are within the molecular weight range of a fragment formed by residues 850-930. It is unclear why this banding loss occurred only in S908A and dephosphorylated Ycf1. We initially attributed this effect to increased susceptibility to protease. In light of our cryo-EM structure, we interpret this as a protective effect where the dephosphorylated R-domain resides in the substrate cavity and may be shielded from protease. We further conclude from the similarities in the digest patterns of dephosphorylated and S908A Ycf1 that the observed changes in

ATPase rate and cell survival correlate to changes in R-domain structure/dynamics.

## Discussion

Our dephosphorylated Ycf1 structure reveals an inactive state that offers insights into the phosphoregulation of ABCC transporters by autoinhibition. The dephosphorylated R-domain packs inside the Ycf1 substrate cavity to lock the transporter in a conformation similar to IFwide in the phosphorylated state[5,19]. In this state, phosphorylation of several adjacent residues along the R-domain helix, specifically S903, S908, and T911, is likely the master switch that controls the transition of the R-domain from an autoinhibitor to activator conformation. In this context, a few mechanisms by which autoinhibition in Ycf1 occurs are likely: 1) the R-domain occludes substrate binding directly to prevent transport, 2) the R-domain physically blocks the assembly of the TMDs and, consequently, NBD dimerization forming the power stroke of the ABC transporter conformational switch, 3) the R-domain blocks engagement of the GRD motif, a key motif required

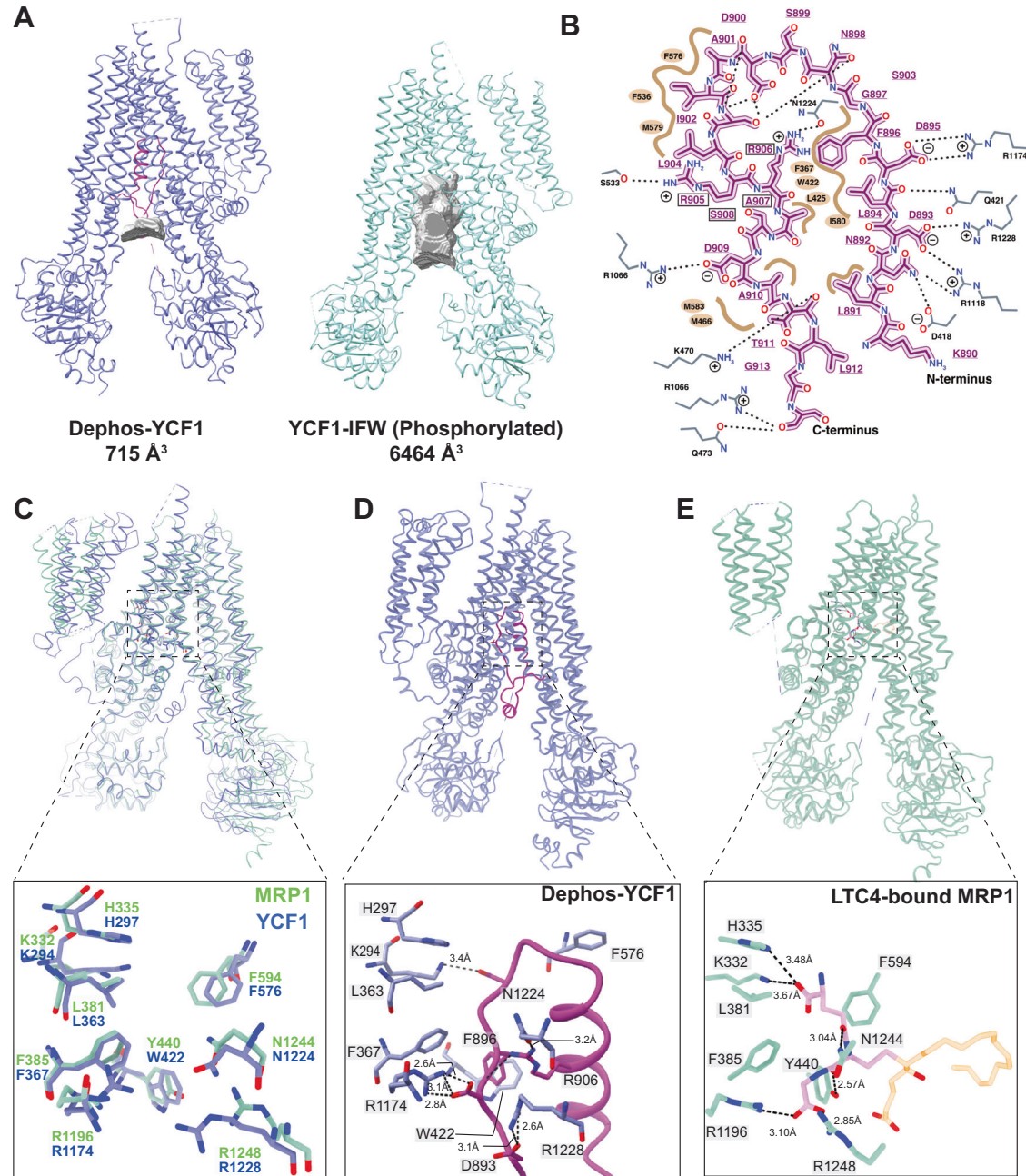

**Fig. 4 | Comparison of transport substrate binding pocket in dephosphorylated Ycf1, phosphorylated Ycf1, and substrate bound MRP1. A** Solvent accessible space-filling model of the Ycf1 substrate cavity. **B** Ligplot detailing specific interactions between dephosphorylated Ycf1 and the R-domain in its binding cavity. **C** Comparisons of the phosphorylated Ycf1 R-domain binding site to MRP1 (PDB IB: 5UJA[23]). **D** Detailed view of the dephosphorylated Ycf1 R-domain interaction in the substrate binding pocket. **E** Interaction network of the glutathione moiety in Leukotriene C4.

for allosteric coupling between NBDs and TMDs[25], and lastly 4) R-domain sequesters away from the allosteric activation site on NBD1, which our previous phosphorylated Ycf1 structure[5] showed requires R-domain docking for Ycf1 stimulation. Scenarios 1-3 suggest a direct inhibition of function by the dephosphorylated R-domain and suggest phosphorylation plays a permissive role in transport, whereas scenario 4 represents the loss of a stimulatory function upon losing the R-domain phosphosites when dephosphorylated and suggests a stimulatory role. It is likely that all four scenarios contribute to the autoinhibitory mechanism.

Our findings, therefore, suggest a plausible R-domain activation and deactivation cycle (Fig. 6). In the resting state, the R-domain maintains the transporter in an inactive conformation. Several groups

have predicted this widespread autoinhibitory mechanism to arise from the loss of NBD dimerization by direct binding of the R-domain[18]. Alternatively, some have suggested a role of the R-domain in the blockage of the substrate pore[8,10,22,27]. Our structure, supported by our functional data, shows that the R-domain impacts both functions. However, we still do not know the order of events leading to subsequent phosphorylation at other R-domain phosphosites. Nonetheless, our data taken together show that the phosphorylated R-domain hairpin is released from its inhibitory site and partially disassembles its tertiary structure while maintaining its secondary structure. The R-domain then engages the potentiating site on NBD1 near the junction of NBD1, ICL3, and the lasso domain where the phosphorylated residues S903, S908, T911, and S914 in the R-domain can

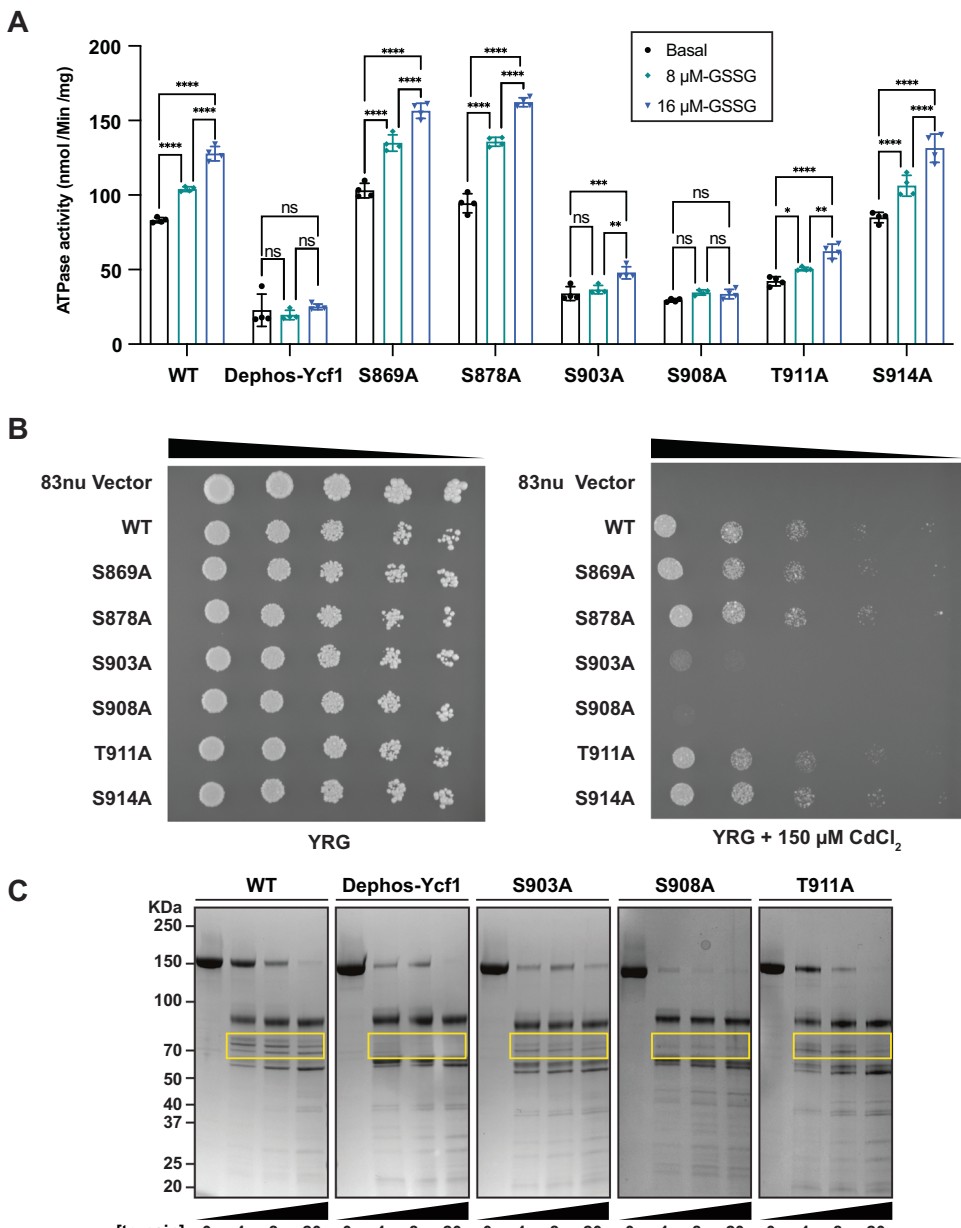

**Fig. 5 | Functional characterization of Ycf1 activity upon dephosphorylation and mutation. A** ATPase rates in different Ycf1 phosphorylation states and Ycf1 mutants. ATPase rates of Ycf1 with and without glutathione acceleration. All values are the mean of four replicates ± standard deviation (S.D.) Source data are provided as a Source Data file. **B** Spot assays showing survival of cells with Ycf1 mutants. First spot of each mutant is 0.1 OD at 600 followed by 5-fold serial dilution as represented by black triangle. **C** Limited proteolysis of phosphorylated Ycf1, dephosphorylated Ycf1, and R-domain mutants.

interact with a cluster of positively charged residues (K615, K617, K655, R716, R770, R206)[5].

This mechanism is consistent with several previous biochemical results from full-length protein investigations. Solvent accessibility experiments on full-length CFTR using tryptophan fluorescence quenching showed a significant (~20% or ~450 residues) change in the solvent-accessible surface area upon phosphorylation consistent with the release of a large domain[20]. In another solvent accessibility experiment using deuterium exchange by attenuated total reflection Fourier transform infrared (ATR-FTIR) spectroscopy, these changes localized to the TMDs[20]. Importantly, these changes are not accompanied by any significant changes in secondary structure. Together, these findings support our hairpin model of transporter autoinhibition in which the unphosphorylated R-domain is buried in the substrate cavity and then completely

unwinds the 902-912 helix upon phosphorylation to bind NBD1 with an altered secondary structure.

Within the R-domain hairpin bundle, our data suggest that S908 phosphorylation is the most important to relieve autoinhibition. When mutated to alanine, S908A Ycf1 most closely mimics the dephosphorylated Ycf1 state and, as shown by us and others[5,6], is the most functionally altered variant compared to wild-type Ycf1. Our data also indicate that apart from S908, several R-domain phosphorylation sites only partially affect the ATPase activity and protease accessibility. Kinetic experiments using light activatable serines[28] support these findings by showing that S813 (equivalent to S908 in Ycf1) is rate-limiting for CFTR activation. Patch clamp measurements in CFTR give further evidence of the importance of this site[27].

The exact sequence of events that present S908 to kinases remains a mystery in the context of our structure because S908 is not

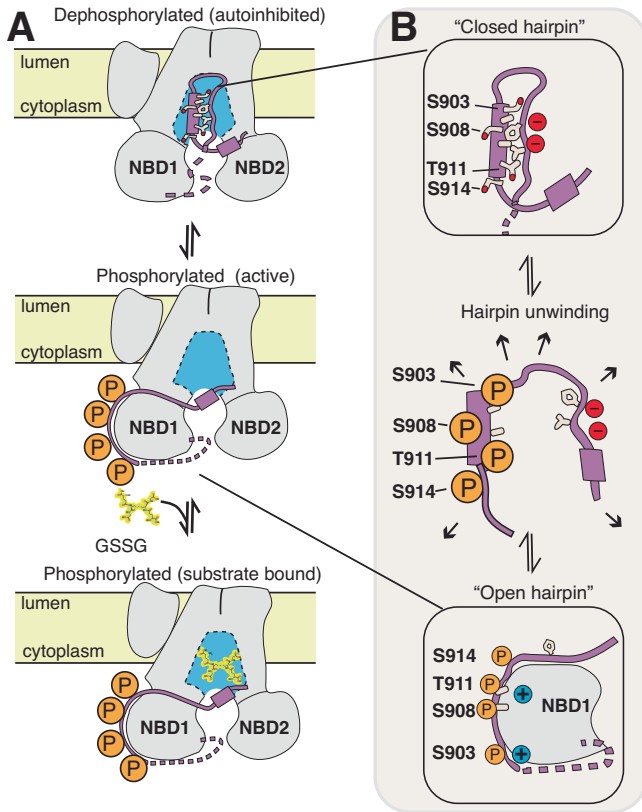

**Fig. 6 | Proposed model for R-domain autoinhibition and activation. A** Overall cycle of Ycf1 transitioning from a dephosphorylated state to a phosphorylated state. The R-domain is colored purple, and the substrate cavity is colored blue. **B** A closeup of the R-domain hairpin loop (purple) is shown with interacting hydrophobic residues (tan).

accessible in the buried state we observe. S908 accessibility could be a stochastic process or could be driven by the phosphorylation of a secondary site, helping explain the partial phenotypes of other phosphorylation sites. Recent data on CFTR phosphoregulation shows that the dephosphorylated R-domain indeed can stochastically release to engage PKA, lending support to this mechanism[29]. Upon phosphorylation along sites S903, T911, and S914, and critically S908, it is likely that once a critical mass of phosphorylation is achieved in such a small volume, electrostatic repulsion disfavors helix formation and destabilizes the bound hairpin (Fig. 6). Instead, the unwound phosphorylated state of the R-domain now favors engagement with a positively charged surface of NBD1, consistent with phosphomimetic mutation experiments suggesting negative charge as sufficient to cause for R-domain rearrangement[30]. These results also match NMR results in CFTR that show a loss of helical propensity in the R-domain C-terminus upon phosphorylation[15].

Multiple factors suggest that an activation cycle dependent on phosphorylation/dephosphorylation may be conserved across several ABCC family members. The R-domain C-terminus maintains a higher degree of conservation between the several ABCC members, including CFTR, MRP1, and Ycf1. In members with multiple phosphorylation sites, the relative spacing between phosphosites is also conserved, especially in proximity to the S908 site, which has been considered important in intrinsically disordered domains even when sequence conservation is not high (Figure S3). In Ycf1, this conserved spacing can plausibly be explained by the spacing imparted by the helical nature of the phosphosites in the dephosphorylated structure. Though these transporters maintain different functions, MRP1 and Ycf1 can functionally replace one another, and all three have been reported to have

glutathione transport activity[2,9]. Lastly, during the review of this manuscript, two structures of the homologous human and rat transporter ABCC2 showed a similar configuration of the R-domain in the substrate cavity. However, in these cases, the structure of the R-domain placement was not directly shown to be phosphorylation-dependent and instead was linked to the presence of substrate[31,32], although activity was shown to be dependent on phosphorylation and supports our autoinhibition model.

The substrate cavity interactions in the autoinhibited R-domain conformation are also well-conserved among many ABCC members. The dephosphorylated R-domain makes several contacts synonymous with glutathione binding in MRP1[23]. The R-domain architecture shares a striking resemblance to the IC47 viral peptide inhibitor from herpes simplex virus bound to the TAP1/TAP2 transporter substrate cavity (Figure S4). This inhibitor adopts a hairpin architecture with two helices (compared to one for Ycf1) stabilized by a hydrophobic zipper and electronegative residues within the hairpin loop[26]. Other ABC transporter families with similar linkers and controlled by phosphorylation may have similar mechanisms. ABCB1 (P-glycoprotein) is suggested to have a possible helical hairpin structure in its NBD1-TMD2 linker based on similarity to excreted WXG motif hairpin proteins (Figure S4) that are secretion substrates for Type VII type ABC secretion system[22].

In conclusion, our dephosphorylated Ycf1 structure shows a dramatic rearrangement of the R-domain that occludes the transporter substrate binding pocket, resembling the peptide-bound state of TAP1/TAP2. It suggests a mechanism where the R-domain blocks activity by displacing substrate, by preventing NBD dimerization, and by sequestering allosteric elements that communicate ATP binding to the substrate cavity. Lastly, our data support S908 as a dominant driver of Ycf1 activation with a requirement for S903 and T911 to form a phosphorylated bundle, likely both through the release of the R-domain hairpin inhibition and by direct engagement with NBD1.

## Methods

### Ycf1 expression and purification

The *S. cerevisiae* Ycf1 expression construct used here is the same as in our previous study. R-domain phosphorylation site mutants were constructed by site-directed mutagenesis using primers listed in Supplementary Table 1 and verified by sequencing (Elim Biopharmaceuticals, Inc). Protein was expressed and purified as previously described[5]. Briefly, the DSY-5[33] strain of *S. cerevisiae* (Genotype MATa leu2 trp1 ura3-52 his3 pep4 prb1) was transformed with wild-type or mutant Ycf1-encoding plasmids and selected on YNB-His agar medium plates. Transformants were inoculated into a primary culture of 50 mLYNB-His media (yeast nitrogen base with ammonium sulfate 0.67% w/v, glucose 2% w/v, CSM-His 0.077% w/v) and grown for 24 hours at 30 °C with 200 rpm shaking. Then, 15 mL of the primary culture was used to inoculate 750 mL of a secondary YNB-His culture and grown similarly to the primary culture for an additional 24 hours. Protein expression was induced by adding 250 mL of 4X YPG media (4% w/v yeast extract, 16% w/v peptone, and 8% w/v galactose). The culture was grown for an additional 16 hours. Finally, cells were collected by centrifugation at 5000 x g and 4 °C for 30 minutes and stored at −80 °C.

Crude membranes were prepared by cells resuspended in lysis buffer (50 mM Tris-Cl pH 8.0, 300 mM NaCl, protease inhibitor cocktail mix) and subject to bead beating as previously described[6]. Briefly, the lysate was separated from beads by filtering through a coffee filter, and membranes were harvested by ultracentrifugation of the clarified lysate at 112,967xg for 1.5 hours before storage at −80 °C. Membranes were solubilized for 4 hours at 4 °C at a ratio of 15 mL resuspension buffer (50 mM Tris-Cl pH 7.0, 300 mM NaCl, 0.5% 2,2-didecylpropane-1,3-bis-β-D-maltopyranoside (LMNG)/0.05% cholesteryl hemisuccinate (CHS) supplemented with protease inhibitor cocktail) per gram of membrane. The insoluble fraction was separated by centrifugation at

34,155xg for 30 min at 4 °C followed by filtration of the supernatant through a 0.4 μM filter prior to loading onto a 5 mL Ni-NTA immobilized metal affinity chromatography (IMAC) column (Bio-Rad) equilibrated in buffer A (50 mM Tris-Cl, 300 mM NaCl, 0.01% LMNG/0.001% CHS, pH 7.0).

IMAC immobilized protein was purified by first washing with 50 mL (10 column volume (CV)) of buffer A followed by a gradient of buffer B (50 mM Tris-Cl, 300 mM NaCl, 500 mM Imidazole 0.01% LMNG/0.001% CHS, pH 7.0) using the following steps sizes: 10 CV of 6% Buffer B, 2 CV of 10% Buffer B, 2 CV of 16% Buffer B and 2 CV mL of 24% Buffer B. Protein was eluted with 4-5CV of 60% buffer B and diluted immediately with 10-fold buffer A prior to being concentrated at 3095xg at 4 °C in a 100 kDa cutoff Amicon concentrator (Millipore). The process was repeated three times to remove the excess imidazole prior to sample injection onto a size exclusion Superose 6 Increase 10/300 GL column (GE Healthcare) equilibrated in SEC buffer (50 mM Tris, 300 mM NaCl, pH 7.0) supplemented with 0.01% LMNG/0.001% CHS (Figure S5). Protein concentration was measured by Bicinchoninic acid assay (Pierce) and used immediately, without freezing, for biochemical and structural analyses.

Ycf1 dephosphorylation was performed using our previously established method[5]. SEC-purified sample Ycf1/Ycf1-E1435Q (5 μg) was treated with 1 μL of Lambda phosphatase (Lambda PP, NEB) for 1 hr at 30 °C. Following treatment, the sample was subjected to a second round of SEC purification, the same as above, to remove the phosphatase. For cryo-EM samples, SEC purification was performed in SEC buffer containing 0.06% digitonin, concentrated, quantified, and immediately used for grid freezing.

## ATPase activity assay
The ATP hydrolysis activity of purified Ycf1 was measured at 30 °C using the enzyme-coupled assay as described previously[5]. Reaction samples were 75 μL each and contained 6.45 μg of purified protein in a reaction mixture consisting of 20 mM Tris-HCl pH 7.0, 10 mM MgCl₂, 1 mM PEP, 55.7 / 78.0 U/mL PK/LDH, 0.3 mg/mL NADH and 1 mM ATP. Substrate-induced ATPase activity was measured by adding 8 or 16 μM GSSG to the reaction mix. ATPase rates were calculated as a function of NADH consumption measured as the change in absorbance at 340 nM for 45 min on a Synergy Neo2 Multi-mode Microplate Reader (BioTek). ATPase rates were calculated using linear regression in GraphPad Prism 9.

## Yeast survival spot assay
A Ycf1-KO *S. cerevisiae* strain transformed with wild-type or mutant Ycf1 was grown overnight on YNB-His agar plates. The following day, a dilution of the overnight culture with an OD600 reading of 0.1 was prepared in 0.9% saline. Five-fold serial dilutions were prepared and 5 μL of each spotted onto YRG (yeast nitrogen base with 0.67% w/v ammonium sulfate, 1% w/v raffinose, 2% w/v galactose, 0.077% w/v CSM-His, and 2% w/v agar) agar plates with and without 150 μM CdCl₂. Plates were incubated for five days at 30 °C and photographed using a Bio-Rad Chemidoc MP Imaging System. The assay was performed in quadruplets (two biological replicates, with two technical replicates per biological replicate). Representative images are shown in Fig. 4.

## Limited trypsin proteolysis
Ycf1 samples were treated with trypsin from bovine pancreas (Sigma) at different concentrations (0, 4, 8, and 20 μg/mL) for 1 h on ice in a 20 μL reaction volume containing 6 μg of purified wild-type or mutant Ycf1. After 1 h, the reaction was stopped by adding 1 μL of soybean trypsin inhibitor (Sigma) from a stock at 1 μg/mL concentration and incubating for an additional 15 min on ice. Gel loading samples were prepared by mixing 5 μL of each reaction with 1X SDS loading dye containing 100 mM DTT and separated on 4−20% SDS-PAGE gels (Bio-

Rad) before staining with Coomassie Brilliant Blue R-250 stain. Experiments were performed at least in triplicate.

## Dephosphorylated Ycf1 grid preparation and cryo-EM data collection
5 μL of dephosphorylated Ycf1-E1435Q at 10.56 mg/mL were applied on QF-1.2/1.3-4Au grid (Electron Microscopy Sciences) using a Leica EM GP2 set to 10 °C and 80% humidity. Following sample application, the grid was incubated for 10 s followed by a 3.5 second blot on Whatman 1 paper and immediately plunge-frozen in liquid ethane equilibrated to −185 °C.

A total of 5,904 movies were recorded on a Titan Krios at 300 kV equipped with a K3 Summit detector (Gatan) using SerialEM v4.0.0beta software at the Pacific Northwest Center for Cryo-EM (PNCC). Data were collected in super-resolution mode at 81,000X magnification with a defocus range of −0.6 to −1.9 μm. Each movie contains 75 frames with a total electron dose of ~52 electrons /Å².

## Cryo-EM data processing
The dephosphorylated Ycf1-E1435Q dataset was processed starting with Relion 4.0[34]. Image stacks with a pixel size of 1.0695 Å/pixel were generated from drift correction using MotionCor2[35] in Relion4.0 followed by contrast transfer function (CTF) estimation using CTFFIND4.1[36] (Fig. 2SA). Micrographs with estimated CTF resolutions of 4.5 Å or better were selected (5,152 micrographs) and subject to template-free manual particle picking and 2D classification (Fig. 2SB). These 2D classes were then used as templates for automated particle picking, and a total of 5,424,196 particles were picked and extracted with 4X binning to 4.278 Å/pixel and a box size of 240 pixels (Figure S1A). Three rounds of 2D classification were performed, resulting in 1,341,751 particles subjected to ab-initio 3D map generation. Using the ab-initio 3D map as a reference, two rounds of 3D classification were performed. The two best classes were selected from the second round of 3D classification. Particles were unbinned to a pixel size of 1.0695 Å/pixel and extracted with a 240-pixel size box, further refined, and then re-extracted to a box size of 360 pixels at a size of 1.0695 Å/pixels. 3D refinement was then performed in RELION using SIDE-SPLITTER and normal 3D refine with iterative rounds of CTF refinement, Bayesian polishing, and postprocessing in RELION resulting in a 3.13 Å resolution map and a 3.56 Å resolution map.

From Relion, the 2 classes were imported into Cryosparc for improvement with Non-Uniform refinement and local refinement. 3D variability analysis on the best class showed that considerable heterogeneity remained with stronger R-domain density associated with a closer arrangement of NBDs. Both classes were recombined and refined in Cryosparc to 2.87 Å, then subject to 3D classification with 5 classes. The two smallest classes showed the strongest R-domain density. These were then combined and refined to an ultimate resolution of 3.18 Å. 3D flex refinement[37] was finally used with training resolution limited below the reconstruction resolution (4.27 Å) as recommended by the software programmers to limit bias. Reconstruction and refinement led to a final map of 3.11 Å with greatly improved connectivity and signal from the R-domain that enabled tracing of the model from residue 890 through the entire C-terminus of the R-domain.

## Ycf1 model building and manuscript preparation
A Ycf1 model from AlphaFold2[38] (accessed 7-22-2021 with Uniprot ID P39109) was used as an initial model combined with elements of our previous Ycf1 structure (PDBID: 7M69)[5]. The Isolde plugin for ChimeraX[39] and manual fitting in COOT[40] were used for model improvement. Iterative cycles of real-space refinement were performed by Phenix[41] against the final cryosparc map. B-factor blurred (i.e. positive B-factor applied) Relion maps, were also generated in COOT as well as Phenix Density Modification[42],DeepEMhancer[43], and

EMReady[44] in earlier stages of model building. Modelangelo was used for interpretation of the R-domain (though not used for the final model), with both fasta file and no fasta file modes used to obtain an unbiased interpretation of our density[45]. All statistics are reported against the Relion Postprocess map (Supplementary Table 2). Figures were prepared using UCSF ChimeraX. R-domain and ligand binding analysis was performed using the software Ligplot[46]. Finally, in reorganizing and re-outlining portions of the manuscript, GPT3.5 (https://chat.openai.com/) was used for suggestions of organization. However, none of the text output was used in the manuscript text.

### Reporting summary

Further information on research design is available in the Nature Portfolio Reporting Summary linked to this article.

## Data availability

The cryo-EM density map data generated in this study have been deposited in the Electron Microscopy Data Bank (EMDB) database under accession code EMD-40451. The structure model data used in this study are available in the Protein Data Bank (PDB) database under accession code 8SG4 and have been deposited to figshare available at https://doi.org/10.6084/m9.figshare.25284613. Source data are provided with this paper.

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

## Acknowledgements

A portion of this research was supported by NIH grant U24GM129547 and performed at PNCC at OHSU and accessed through EMSL (grid.436923.9), a DOE Office of Science User Facility sponsored by the Office of Biological and Environmental Research. This work was also supported by grants from the National Institute of Allergy and Infectious Disease (NIH R01 AI156270 (TMT)), NIH S10 OD011981 (Life Sciences North Imaging Facility at the University of Arizona) and the University of Arizona BIO5 Postdoctoral Fellowship Award (NKK). We also thank the staff at the Pacific Northwest Center for Cryo-EM (PNCC), especially Nancy Meyer, for assistance with data collection. We also thank Tarjani M. Thaker for critical reading of the manuscript, Meghna Gupta for assistance with data interpretation, and members of the Tomasiak lab for helpful discussions.

## Author contributions

Conceptualization: NKK, TMT; Methodology: NKK, TMT; Investigation: NKK, TMT; Visualization: NKK, TMT; Funding acquisition: TMT; Supervision: TMT; Writing – original draft: NKK, TMT; Writing – review & editing: NKK, TMT.

## Competing interests

The authors declare no competing interests.
