## [Peer Review File · Nature Communications]

Structural Basis for Autoinhibition by the Dephosphorylated Regulatory Domain of Ycf1REVIEWER COMMENTS

Reviewer #1 (Remarks to the Author):

In this contribution, Khandelwal & Tomasiak present the full-length structure of Ycf1 ABC-transporter in its dephosphorylated state and based on the comparison with the previously resolved structure(s) they propose the model for R-domain auto-inhibition and activation. To the great disappointment after the inspection of provided model and EM density I am not convinced with the modelling of R-domain (the rest of the model looks good). For R-domain the density is extremely weak / absent, hence it is impossible to unambiguously assign the sequence in this region and moreover even to trace the main chain. For example at the position where the authors tried to model a hairpin (where important phosphosites are located S903, S908, T911) - it is impossible to place 8 missing residues, since there is simply not enough space (and I did try to model it myself) which clearly indicates the sequence assignment here is not correct as the sequence must be shifted. Hence all the conclusions based on the current assignment / modelling of R-domain are dubious.

Another question I have is about the processing of EM data: the first class after 2nd 3D classification resembles IF_{narrow} conformation - did the authors try to process it any further? Also what is the difference between 2nd and 3rd class in the same 3D classification round? (29 and 46%) Why didn't the authors try to process the 2nd class? (the resolution is not that different).

In addition, it is also essential to show SDS-gel to estimate the expression levels of mutants (in addition to SEC profiles in S.Fig. 5)

Minor issues:

- please use greek letters where necessary, e.g. γ -hydroxyl instead of g-hydroxyl
- please separate values and units, e.g. 3.1Å should be 3.1 Å etc., 50mM should be 50 mM, etc.
- in Results the authors say electron density, but I assume they only discuss EM structures, so electron density is incorrect.
- area should be squared, e.g. 1800 Å² instead of 1800Å
- In Figs 1-2 the lipid densities can be removed as they are not really discussed in the manuscript
- Fig 4 legend, a typo in the word 'site'

Reviewer #2 (Remarks to the Author):

This study presents the first high-resolution cryo-EM structure of the yeast Ycf1 ABC transporter in its dephosphorylated state. Ycf1 belongs to the ABCC subfamily whose members are characterized by a regulatory (R) region/domain which connects the two ABC-typical NBD-TMD halves. The R domain is subject to phosphorylation and controls activity of several family members. Cryo-EM structures for Ycf1 have so far been solved only in the phosphorylated state. In phospho-Ycf1 the R domain occupies a peripheral position, associating with the external surface of NBD1. The position of the dephospho-R domain and the mechanism by which it inhibits Ycf1 activity are unknown. The present structure provides a detailed view of the dephospho-R domain and explains the mechanism of its inhibitory action.

Not being a structural biologist I am not qualified to judge the quality of the data, but comparing the EM density map with the model it seems to me that the R-domain corresponds to one of the less well defined regions of the map. Since it is exactly this part that forms the focus of the study, the reliability of the model should be verified by a structural biologist expert reviewer. Assuming that the model is valid, the conclusions drawn are logical and provide a substantial advance in our understanding of Ycf1 regulation.

Below are some suggestions for improving the clarity of the presentation.

1.) "Here, we sought to determine the structural changes in dephosphorylated Ycf1-E1435Q"

Explain the role of residue E1435 and the rationale for using the E-to-Q mutant.

2.) "~1800Å of solvent-exposed surface area"

Angstrom should be squared.

3.) "...that includes Lys294 and Gln432. Finally, Thr914 forms a pair of interactions"

Thr 911, not 914.

4.) "dephosphorylated R-domain contacts with a critical allosteric junction, the "GRD" motif

of ICL-1, which couples with the X-loop of NBD1 to communicate ATP binding"

Please show a figure which illustrates this interaction.

5.) "The ICP47 peptide exhibits an identical architecture"

I would not call the helix/helix hairpin of ICP47 and the helix/strand hairpin of the Ycf1 R domain "identical structures". In Suppl. Fig. 4 maybe show an image of the full structure to highlight the fact that ICP47 also binds into the substrate cavity of TAP1/TAP2.

6.) "phosphosites S903 and S908, which severely impact ATPase activity when mutated, could not be resolved in the dephosphorylated cryo-EM map. These results suggest that either 1) glutathione binding is blocked or 2) glutathione binding is not allosterically communicated with the ATPase sites in these mutants"

I don't understand this inference. Please explain why these results lead to the above conclusions.

7.) "phosphorylation site mutants S869A, S876A,..."

Correctly S878A?

8.) "showed no (or slightly elevated) survival"

Do you mean no reduction (or slight elevation) of survival?

9.) "Scenarios 1-3 are antagonistic, whereas scenario 4 represents the loss of a stimulatory effect but not function."

The meaning of this sentence is unclear.

10.) "Finally, these experiments show the opposite effect on isolated NBD1"

This sentence is unclear. Which experiments? Opposite to what?

11.) "Patch clamp measurements support this result in an R-domain fragment of CFTR containing S813 (Ref. 25) (equivalent to S908 in Ycf1) and S798"

PMID:36695813 is a much more relevant study on the role of S813 in the context of full-length CFTR.

12.) "The exact sequence of events exposing S908 to kinases remains a mystery... because S908 is not accessible to kinases in the bound state... S908 accessibility could be a stochastic process..."

For CFTR there is evidence for a stochastic process, with release preceding phosphorylation (PMID: 32817533).

13.) "Lastly, our data support S908 as a dominant driver of Ycf1 activation... by direct stimulation of NBD1."

What is the meaning of the last part of this statement? What exactly is stimulated in NBD1?

14.) Fig. 3A: The coloring (magenta vs. pale pink) of the two helices 890-899 and 920-935 is reversed between the left and the right panel. This is confusing for the reader. Please use a consistent coloring scheme.

15.) Fig. 4C-D-E: The full transporter structures are so pale that they are virtually invisible.

16.) Figure 5B: What do the individual columns of spots represent? What is the meaning of the black triangles above the images? Please explain.

17.) Legend for Figure S3: "Asterix (*) denotes sites to be confirmed to be phosphorylated"
For CFTR the asterisks denote exactly the sites that have already been confirmed to be phosphorylated.

We thank the reviewers for evaluating our manuscript “Structural Basis for Autoinhibition by the Dephosphorylated Regulatory Domain of Ycf1”. We appreciate the positive comments reflecting the importance of the auto-inhibited state we have identified. We also note the main criticisms, which primarily concern the modeling of the R-domain in our cryo-EM maps. Upon reanalysis of our maps and model, we agree with these criticisms and have rebuilt and reanalyzed our model and cryo-EM data to address reviewer #1’s comments. In this letter, we begin with a discussion of these larger criticisms in the discussion below, then follow with a discussion of how we addressed other issues raised in the review of our manuscript.

To summarize our detailed response to the comments concerning the modeling of the R-domain, we reprocessed the dataset in a different software package, CryoSPARC¹. This approach was used to identify any remaining heterogeneity leading to weak density in the R-domain region we had originally modeled. We have now successfully isolated a more homogenous class with a more evident electron potential density corresponding to the R-domain, allowing us to address this comment as described below. To facilitate our manuscript review, the PDB file and cryo-EM map are included with our resubmission files.

Reviewer comments

Response to Reviewer #1

“To the great disappointment after the inspection of provided model and EM density I am not convinced with the modelling of R-domain (the rest of the model looks good). For R-domain the density is extremely weak / absent, hence it is impossible to unambiguously assign the sequence in this region and moreover even to trace the main chain. For example at the position where the authors tried to model a hairpin (where important phosphosites are located S903, S908, T911) - it is impossible to place 8 missing residues, since there is simply not enough space (and I did try to model it myself) which clearly indicates the sequence assignment here is not correct as the sequence must be shifted. Hence all the conclusions based on the current assignment / modelling of R-domain are dubious.”

After carefully reanalyzing our data following the reviewer's comments, we agree with this interpretation of our R-domain density and thank them for carefully analyzing our data. We agree that the density was weak in this region and that the missing region was challenging to fit in the remaining cavity. We reexamined our data with new tools in CryoSPARC and found residual heterogeneity that limited the R-domain interpretability. Notably, we used the 3D Variability analysis recently added in Cryosparc and found that this class contains a substantial fraction of particles with Nucleotide Binding Domains (NBDs) closer together (**Fig. 1**). Notably, the stronger and more connected R-domain density came with this conformation. Realizing that this class still contained dissimilar particles, we sought to use CryoSPARC tools to better separate particles from this class to improve the reconstruction. We also combined particles from the other 2nd Relion class since they matched this second conformation with the rationale that they may also have particles with stronger R-domain density.

We then fed this class into a 3D classification job in CryoSPARC with five classes. We found results matching the previous 3D variability job with the two least populated classes containing the strongest R-domain density and the closer NBDs. Combining particles from these classes, we found that the resulting model had exceptionally strong density for the R-domain. After several additional rounds of processing, we subjected our best model to an additional round of refinement using 3Dflex in CryoSPARC to evaluate map heterogeneity and further improve weak regions in

the consensus reconstruction, which 3Dflex has been shown to do exceptionally well². This refinement resulted in maps with similar resolution to our previous model but a significant gain in connectivity that enabled the building of the entire C-terminal half of the R-domain.

Figure 1. CryoSPARC 3D Variability Analysis of dephosphorylated structure from previous round of review. **A.** Cryo-EM map of the most farthest distance NBD arrangement. This state is the dominant class and resembles the structure we submitted in the previous round of review. **B.** The new state with NBDs closer together. This state formed the basis for the new updated structure. **C.** A comparison of both states.

The strong density now clearly revealed a topology consistent with 1) a helical segment of approximately three turns or 10-12 amino acids, 2) several bulky residues including two well-resolved arginines that are a central part of a canonical phosphorylation domain (905-908; RRAS), and 3) how the R-domain connects to TMD2. Using these features as landmarks, we rebuilt the R-domain in a new conformation with the opposite direction of the disjointed R-domain from our previously submitted dephosphorylated structure. We verified this new model in 3 complementary ways.

1. We verified the fit of our R-domain model using the Q-score³, a recommended metric for cryo-EM map interpretation. Our overall Q-score was 0.58, which is higher than the recommended value of 0.56 for our resolution (3.1Å). We also found that the R-domain density Q-score in this region was now 0.53, which is near the recommended value and consistent with the strong density in this region.
2. We used the new automated model-building package Model Angelo⁴, which is distributed in the new release of Relion (Relion5 – one of the major software packages for cryo-EM data processing) as an unbiased metric to judge our model of the R-domain. This software combines a deep-learning-based model-building approach with an algorithm for interpreting electron potential density with accuracy in model building that is shown to be near-expert⁴. Although we present our manually assigned model in our manuscript, we use Model Angelo as an independent assessment of our model since it is unbiased owing to the lack of a structure as an input. We tested both available modes by providing a fasta file of Ycf1 and in cryo-EM map-only mode (i.e., no fasta file is provided). In both cases, it predicted a similar interpretation of the structure with a helical density in the R-domain region 901-912 pointing down from the substrate cavity toward the NBDs. The sequence-free mode predicted that this region should contain a sequence very similar to the RRAS sequence (shown below – **Fig. 2**) and outputs a Hidden Markov Model of the predicted sequence in this region. This closely matches the sequence that we independently modeled.

3. Independently, we verified the secondary structure using NMR (published by the Forman-Kay group on the related protein CFTR). This assignment agrees closely with our model⁵ and include this analysis in an updated supplementary figure 2.

Figure 2. Comparison of Ycf1-dephos structure with Model Angelo automatic build.
A. New model of R-domain from 890-920 **B.** Model Angelo automated building (raw cif) using an input fasta sequence **C.** Model Angelo automated building with no input fasta sequence provided. Note that although the algorithm built only fragments in the no-sequence case, it matches both the direction of our model and the location of several prominent amino acids, notably R905 and R906.

Again, we thank the reviewer and believe their suggestions have greatly improved our interpretation of the data.

Finally, reviewer #1 had comments regarding the original classes from Relion.

Another question I have is about the processing of EM data: the first class after 2nd 3D classification resembles IF_{narrow} conformation - did the authors try to process it any further?

These comments are now addressed with our new map described above. This additional class the reviewer describes (first class after 2nd 3D classification) was combined with our other (3rd class) class after importing to CryoSPARC in our new workflow (Supplementary Fig 1). This class had a closer NBD association and resembled our IF_{wide} conformation (our previous class (3rd class) had a wider separation of NBDs and thus was different than IF_{narrow} or IF_{wide}). However, the reconstruction suffered originally from extra density from nearby particles around the NBDs. We used particle subtraction in Cryosparc to remove this density and found that this class could be combined with our previous class. In CryoSPARC, the heterogeneity processing tools recently made available allowed us to extract useful particles from both classes and develop the reconstruction used here.

Also what is the difference between 2nd and 3rd class in the same 3D classification round? (29 and 46%) Why didn't the authors try to process the 2nd class? (the resolution is not that different).

In this version, we have now combined these particles and reclassified them. The remaining unused class has a much weaker R-domain density.

In addition, it is also essential to show SDS-gel to estimate the expression levels of mutants (in addition to SEC profiles in S.Fig. 5)

We have now added gels for the mutants (S869A, S878A, and S914A) in supplementary figure S5. Our main text, Figure 5C, includes the SDS-PAGE gels for the wild-type Ycf1 (WT), dephosphorylated Ycf1 (Dephos), S903A, S908A, and T911A mutants.

Minor comments

- please use greek letters where necessary, e.g. γ -hydroxyl instead of g-hydroxyl

We have corrected the text to include Greek letters where they are missing.

- please separate values and units, e.g. 3.1\AA should be 3.1 \AA etc., 50mM should be 50 mM , etc.

We have corrected all units in the manuscript.

- in Results the authors say electron density, but I assume they only discuss EM structures, so electron density is incorrect.

This was an oversight and electron density has been corrected to electron potential density.

- area should be squared, e.g. 1800 \AA^2 instead of 1800\AA

We have corrected this error.

-In Figs 1-2 the lipid densities can be removed as they are not really discussed in the manuscript

We have removed the lipid densities from both figures.

-Fig 4 legend, a typo in the word 'site'

We have corrected the error in the figure legend.

Response to Reviewer #2

Not being a structural biologist I am not qualified to judge the quality of the data, but comparing the EM density map with the model it seems to me that the R-domain corresponds to one of the less well defined regions of the map. Since it is exactly this part that forms the focus of the study, the reliability of the model should be verified by a structural biologist expert reviewer. Assuming that the model is valid, the conclusions drawn are logical and provide a substantial advance in our understanding of Ycf1 regulation.

We thank the reviewer for their positive assessment of our work and for their helpful comments.

1.) *"Here, we sought to determine the structural changes in dephosphorylated Ycf1-E1435Q" Explain the role of residue E1435 and the rationale for using the E-to-Q mutant.*

We have added a rationale for the E1435Q mutant with several references to previous structures where a similar mutant was used in this section on page 3.

2.) *"~1800Å of solvent-exposed surface area" Angstrom should be squared.*

We have corrected this error.

3.) *"...that includes Lys294 and Gln432. Finally, Thr914 forms a pair of interactions" Thr 911, not 914.*

With the new structure of the R-domain, these interactions have been changed on page 4.

4.) *"dephosphorylated R-domain contacts with a critical allosteric junction, the "GRD" motif of ICL-1, which couples with the X-loop of NBD1 to communicate ATP binding" Please show a figure which illustrates this interaction.*

In the revised Figure 3D, the GRD motif interaction between R-domain residue D920 and residue R1058 of the GRD motif is now depicted.

5.) *"The ICP47 peptide exhibits an identical architecture "I would not call the helix/helix hairpin of ICP47 and the helix/strand hairpin of the Ycf1 R domain "identical structures". In Suppl. Fig. 4 maybe show an image of the full structure to highlight the fact that ICP47 also binds into the substrate cavity of TAP1/TAP2.*

We agree and have changed the language to highlight their differences. The revised text on page 4 now reads:

"The ICP47 peptide also exhibits a hairpin-type architecture, with the difference being two helices in ICP47, that displays a highly similar set of contacts throughout the substrate cavity."

An image of the entire structure of TAP1/TAP2 with ICP47 has been added to Suppl. Fig. 4.

6.) *"phosphosites S903 and S908, which severely impact ATPase activity when mutated, could not be resolved in the dephosphorylated cryo-EM map. These results suggest that either 1) glutathione binding is blocked or 2) glutathione binding is not allosterically communicated with the ATPase sites in these mutants" I don't understand this inference. Please explain why these results lead to the above conclusions.*

We agree that this section needs to be clarified. We have modified the text in this section to remove reference to the cryo-EM map. We now simply state that lack of GSSG-dependent ATPase stimulation leads us to conclude that either GSSG does not bind or does bind but does not allosterically couple to the NBDs on page 5.

7.) *"phosphorylation site mutants S869A, S876A,..." Correctly S878A?*

Thank you for catching this error. We have changed it in the text.

8.) *"showed no (or slightly elevated) survival"*
Do you mean no reduction (or slight elevation) of survival?

Yes, the reviewer is correct; we have corrected the text to fix this error.

9.) *"Scenarios 1-3 are antagonistic, whereas scenario 4 represents the loss of a stimulatory effect but not function."*
The meaning of this sentence is unclear.

We agree that this statement needs to be clarified. We have simplified it to directly contrast the various stimulatory or inhibitory roles of different R-domain states. We now state on page 6:

"Scenarios 1-3 suggest a direct inhibition of function by the dephosphorylated R-domain, whereas scenario 4 represents loss of a stimulatory function upon losing the R-domain phosphosites when dephosphorylated."

10.) *"Finally, these experiments show the opposite effect on isolated NBD1".*
This sentence is unclear. Which experiments? Opposite to what?

We have removed this reference to isolated NBD1 experiments. Our new results and interpretation of the R-domain now almost exactly match the findings from the NMR work on isolated NBD1 that is mentioned⁵. Since our structure is a closer match to this work, we felt it unnecessary to keep this comparison.

11.) *"Patch clamp measurements support this result in an R-domain fragment of CFTR containing S813 (Ref. 25) (equivalent to S908 in Ycf1) and S798"*
PMID:36695813 is a much more relevant study on the role of S813 in the context of full-length CFTR.

We have added this reference highlighting the importance of S813 on Page 6, and it is cited as reference 28.

12.) *"The exact sequence of events exposing S908 to kinases remains a mystery... because S908 is not accessible to kinases in the bound state... S908 accessibility could be a stochastic process..."*
For CFTR there is evidence for a stochastic process, with release preceding phosphorylation (PMID: 32817533).

We thank the reviewer for this suggestion. This reference is indeed a perfect complement to our study. We have added this on Page 7, and it is cited in our revised manuscript as reference 29.

13.) *"Lastly, our data support S908 as a dominant driver of Ycf1 activation... by direct stimulation of NBD1."*
What is the meaning of the last part of this statement? What exactly is stimulated in NBD1?

We have changed the text from stimulate to engage to read:

“likely both through the release of the R-domain hairpin inhibition and by direct engagement with NBD1.”

14.) *Fig. 3A: The coloring (magenta vs. pale pink) of the two helices 890-899 and 920-935 is reversed between the left and the right panel. This is confusing for the reader. Please use a consistent coloring scheme.*

We have fixed the color scheme to be more consistent.

15.) *Fig. 4C-D-E: The full transporter structures are so pale that they are virtually invisible.*

This has been fixed in this version of the manuscript.

16.) *Figure 5B: What do the individual columns of spots represent? What is the meaning of the black triangles above the images? Please explain.*

We have added an explanation in the figure legends to describe the figure. The added text reads.

Figure legends Figure 5 (B)

“Spot assays showing survival of cells with Ycf1 mutants. First spot of each mutant is 0.1 at OD at 600 followed by 5-fold serial dilution as represented by black triangle.”

17.) *Legend for Figure S3: "Asterix (*) denotes sites to be confirmed to be phosphorylated" For CFTR the asterisks denote exactly the sites that have already been confirmed to be phosphorylated.*

The text has been modified to fix this typo.

REFERENCES

1. Punjani, A., Rubinstein, J. L., Fleet, D. J. & Brubaker, M. A. cryoSPARC: algorithms for rapid unsupervised cryo-EM structure determination. *Nat Methods* **14**, 290–296 (2017).
2. Punjani, A. & Fleet, D. J. 3DFlex: determining structure and motion of flexible proteins from cryo-EM. *Nat Methods* **20**, 860–870 (2023).
3. Pintilie, G. *et al.* Measurement of atom resolvability in cryo-EM maps with Q-scores. *Nat Methods* **17**, 328–334 (2020).
4. Jamali, K. *et al.* Automated model building and protein identification in cryo-EM maps. 2023.05.16.541002 Preprint at <https://doi.org/10.1101/2023.05.16.541002> (2023).

5. Baker, J. M. R. *et al.* CFTR regulatory region interacts with NBD1 predominantly via multiple transient helices. *Nature Structural & Molecular Biology* **14**, 738–745 (2007).

REVIEWERS' COMMENTS

Reviewer #1 (Remarks to the Author):

I thank the authors for the careful revision and compliment them with the significantly improved manuscript - with the improved density allowing modelling of key elements (such as RRAS motif) the interpretation of obtained results is way more solid and I wholeheartedly recommend it for publication.

A few minor comments: looking at the model I've got an impression that S914 can be modelled differently and most likely is engaged in an interaction with the side chain of Q473. In Fig 1 the lipid densities were removed, but the description is still in the caption. In Fig.2 the lipid densities are still there.

Reviewer #2 (Remarks to the Author):

Based on comments of Reviewer 1 the authors have performed extensive additional structural refinement, which has yielded a higher resolution EM map in the region of the wedged-in R-domain segment of dephospho-Ycf1. As a consequence, the model for the the autoinhibitory segment of the R domain, the major focus of the present study, has been completely rebuilt. I am again not qualified to judge the validity of these additional analyses and the reliability of the final model. These should be verified by Reviewer 1.

All my small suggestions for improving the clarity of the presentation have been implemented. I have no further concerns.

Response to reviewers

We thank the reviewers again for their positive assessment of our work and for their remaining comments. In response to the first reviewer's comments, we have made the suggested changes as described below.

"I thank the authors for the careful revision and compliment them with the significantly improved manuscript - with the improved density allowing modelling of key elements (such as RRAS motif) the interpretation of obtained results is way more solid and I wholeheartedly recommend it for publication."

We thank the reviewer again for their comments and suggestions on our work.

"In Fig 1 the lipid densities were removed, but the description is still in the caption. In Fig.2 the lipid densities are still there."

We have updated these figures in this new version; this was an oversight on our part.

"A few minor comments: looking at the model I've got an impression that S914 can be modelled differently and most likely is engaged in an interaction with the side chain of Q473."

We have reexamined this region and are in agreement with this reviewer. In its new position, we have updated the model per the reviewer's suggestion. The γ -hydroxyl of S914 now makes a new contact with Q473 and maintains its previous contact with Arg1060. We have updated Figures 2,3,4 and supplemental figures S2, and S3 to reflect these changes.